*GigaScience*, 2025, **14**, 1–16

**Technical Note**

# SpatialSNV: A novel method for identifying and analyzing spatially resolved SNVs in tumor microenvironments

Yi Liu [1,2], Fan Zhu[1,2], Xinxing Li[2], Xiangyu Guan[1,2], Yong Hou [1,3], Yu Feng [2], Xuan Dong [2], and Young Li [2,4,*]

[1]College of Life Sciences, University of Chinese Academy of Sciences, Beijing 100049, China
[2]BGI Research, Hangzhou, Zhejiang 310030, China
[3]BGI Research, Shenzhen, Guangdong 518083, China
[4]Key Laboratory of Spatial Omics of Zhejiang Province, Hangzhou 310030, China
*Correspondence address. Young Li, BGI Research, XiHu, Hangzhou, Zhejiang Province, China. Email: liyang13@genomics.cn

## Abstract

**Background:** The dynamics of single-nucleotide variants (SNVs) play a critical role in understanding tumor development, yet their influence on shaping tumor microenvironments remains largely unexplored. Spatial transcriptomics offers an opportunity to map SNVs within the tumor context, potentially uncovering new insights into tumor microenvironment dynamics.

**Results:** This study developed SpatialSNV for identifying effective SNVs across tumor sections using multiple spatial transcriptomics platforms. The analysis revealed that SNVs reflect regional tumor evolutionary traces and extend beyond RNA expression changes. The tumor margins exhibited a distinct mutational profile, with novel SNVs diminishing in a distance-dependent manner from the tumor boundary. These mutations were significantly linked to inflammatory and hypoxic microenvironments. Furthermore, spatially correlated SNV groups were identified, exhibiting distinct spatial patterns and implicating specific roles in tumor–immune system crosstalk. Among these, critical SNVs such as S100A11$^{L40P}$ in colorectal cancer were identified as tumor region–specific mutations. This mutation, located within exonic nonsynonymous regions, may produce neoantigens presented by HLAs, marking it as a potential therapeutic target.

**Conclusions:** SpatialSNV represents a promising framework for unraveling the mechanisms underlying tumor–immune crosstalk within the tumor microenvironment by leveraging spatial transcriptomics and SNV-based tissue domain characterization. This approach is designed to be scalable, integrative, and adaptable, making it accessible to researchers aiming to explore tumor heterogeneity and identify therapeutic targets.

**Keywords:** spatial transcriptome, tumor neoantigens, single-nucleotide variants

## Background

Mutation is one of the hallmarks of tumorigenesis, playing a crucial role in programming tumor pathogenesis [1], which emerges mainly in 3 forms, including single-nucleotide variant (SNV), insertion and deletion (indel), and structural variation (SV). Indels and SVs, such as copy number variants (CNVs), involve long-range genomic alterations that are occasionally acquired during tumorigenesis and remain relatively stable during tumor expansion. SNVs are the most common type of genetic variant, occurring at various loci throughout the genome [2]. Most actionable causal mutations in diseases, including cancers and many other genetic disorders, are SNVs [3]. Consequently, sequencing targeted gene panels has dramatically transformed and facilitated disease diagnosis and the customization of therapeutic strategies over the past decade [4]. SNVs accumulate during tumorigenesis and exhibit intercellular heterogeneity [5]. Therefore, SNVs are theoretically more capable of tracing the dynamics of tumor development. However, creating a comprehensive map of SNVs with high resolution and precision remains challenging.

Single-cell sequencing allows us to investigate tumor heterogeneity at a single-cell resolution and has greatly extended our knowledge about tumor evolution [6]. However, losing tissue context information makes revealing intratumor clonal ecosystems and progression difficult. It is essential to capture tumor mutations *in situ* to address these issues. Recently, spatially resolved sequencing technologies have become powerful tools for uncovering tissue slides' molecular and cellular landscapes, including clonal distribution, genetic variation, and tumor cell evolution [7, 8]. The spatial structure of intratumor CNVs can be inferred using spatial transcriptome and genome sequencing [7, 9]. Probe-based *in situ* sequencing has been employed to illustrate the clonal mutation heterogeneity of tumor tissues [10]. While some studies have restored SNVs to their spatial context [7, 11–13], these works mostly focused on the clonal evolution of tumors. There remains a lack of integration across multiple platforms, standardized data formats, and, more importantly, a comprehensive understanding of how spatial SNVs shape the tumor microenvironment *in situ*.

Here, we developed SpatialSNV for calling and analyzing effective SNVs on tumor tissues using spatial transcriptome data. We normalized the SNV count against the total unique molecular identifier (UMI) counts per spatial spot to mitigate the impact of sequencing depth. By further exploring the spatially distinct SNVs, we demonstrated that SNV patterns are more

suitable for tracing the spatial clonal evolution of tumors. To gain deeper insights into tumor microenvironments, we focused on the tumor margins, a region known for its high genetic and expressive heterogeneity [8, 14, 15]. We found that the mutational burden in this region declined in a distance-dependent manner, associated with the inflammatory and hypoxic conditions in the tumor margin microenvironment. Furthermore, we used Spatial-SNV to identify spatially correlated SNVs, termed SNV groups, and found that SNVs within the same groups exhibited similar spatial distributions. SNV groups may reflect the states of tumor cells and surrounding cells in the microenvironment at the tumor margin. We further observed that critical tumor SNVs, such as tumor driver mutations, exhibit specific spatial patterns within the tissue context. Examination of exonic nonsynonymous SNVs also identified KRAS$^{G12D}$, KRAS$^{G12V}$, and S100A11$^{L40P}$ as tumor region-specific mutations in colorectal cancer (CRC) samples, which may generate neoantigens presented by the human leukocyte antigens (HLAs), representing the potential therapeutic target. Notably, single-cell RNA sequencing (RNA-seq) analysis showed that S100A11 was highly expressed in CRC tumor cells, indicating a potential therapeutic target of S100A11$^{L40P}$. SpatialSNV integrates spatial SNV and transcriptomic data to unveil a more comprehensive genetic map of tumor tissue, enhancing our understanding of how SNVs shape the tumor microenvironment. Moreover, effective SNVs can be used to pinpoint tumor-specific exonic nonsynonymous mutations, providing potential therapeutic targets for developing precision medicine strategies.

## Analyses

### SpatialSNV enables effective SNV calling on spatial transcriptomics data

To investigate SNVs within tumors *in situ*, SpatialSNV utilized Mutect2 [16] to call somatic mutations, using panel of normal (PON) germline resource from the GATK resource bundle to filter out potential germline mutations (Methods, Fig. 1A). At the same time, we tested the performance of spatialSNV on multiple spatial transcriptomics platforms, including Stereo-seq [15, 17, 18], Visium [19], Slide-seq [20], and Slide-DNA-seq [9] (Supplementary Fig. S1A). To eliminate false-positive callings caused by low sequencing depth, we examined the number of unique reads supporting each SNV (Supplementary Fig. S1B) and only retained those SNVs supported by more than 20 unique reads. Furthermore, we examined the spatial distribution of SNVs in representative samples, including "DCIS1," "CRC-P19-T," "Slide-RNA-CRC," and "Slide-DNA-CRC" (Fig. 1B). Corresponding spatial transcriptomic profiles were also prepared to assist in the analysis of spatial SNVs (Supplementary Fig. S1C). Most SNVs appeared in only a small number of spots, likely due to technical noise. Spatial-SNV regarded SNVs detected in multiple spots as effective SNVs, which might indicate similar evolutionary progress in tissue. Following this quality control process, we found that Stereo-seq had a relatively high number of effective SNVs, likely due to its higher sequencing depth. We then assessed the correlation between the number of SNV types and the number of unique reads per sample. As anticipated, datasets with higher sequencing depths generally resulted in more effective SNV callings (Fig. 1C, Supplementary Fig. S1D). Notably, due to the extremely low sequencing depth, SpatialSNV could only recover 23 effective SNVs in Slide-DNA-seq data (Supplementary Fig. S1D, E). Therefore, Slide-DNA-seq data were not included in further analysis. Subsequently, we annotated the genomic locations of these effective SNVs and found that

SNVs identified through spatial transcriptomics predominantly originate from gene body regions, with a considerable proportion in untranslated regions (UTRs) (Fig. 1D, E). Then we compared the spatial distribution of effective SNVs derived from public data on the Stereo-seq platform (Fig. 1F). The high consistency of SNVs among samples from the same patient demonstrates the robust SNV calling pipeline of SpatialSNV. Moreover, the high correlation of SNVs within the same tumor and across different tumors suggests associations with driver mutations during tumorigenesis. Consequently, we conducted a Gene Ontology (GO) term enrichment analysis on genes affected by common SNVs across all samples (Fig. 1G). These SNVs impact molecular functions such as cadherin binding and transcription coregulator activity, which are associated with aberrant transcriptional activity [21]. Then, we observed a strong linear association between SNV raw count and gene expression (Supplementary Fig. S2B). To reduce the influence of gene expression and sequencing depth on SNV quantity, SpatialSNV converted the SNV count matrix to a binary matrix and normalized the SNV counts against the total mRNA UMI captured. With this correction, we obtained distinct spatial profiles between transcriptomics and SNVs (Supplementary Fig. S2A) and a significant correlation decrease between RNA expression and SNV counts (Supplementary Fig. S2B). We then observed significant enrichment of SNV mutations on the tumor regions on tumor sections across different spatial transcriptomics platforms (Fig. 1H).

### SpatialSNV identifies tumorigenesis-associated SNVs

Gene mutations play a crucial role in the onset and progression of tumors. To explore whether SpatialSNV is able to identify essential SNVs in tumorigenesis, we utilized transcriptomics data to divide the tumor section into the tumor (marked by *EPCAM* and other tumor-associated markers), tumor-adjacent margins (marked by *VIM*, *COL1A2*, *PTPRC*), and normal tissues (Fig. 2A, B, Supplementary Fig. S3). Subsequently, we validated the reliability of RNA region clustering using the inferCNV(method) (Supplementary Fig. S4). Using the ductal carcinoma in situ (DCIS) section from the Visium platform and CRC section from the Stereo-seq platform as examples, we calculated the SNV mutation frequency and occurrence on each gene (Fig. 2C). We found the B-cell receptor–related genes to be highly mutated with low occurrence due to the somatic hypermutation progress during the development and maturation of B cells. Mitochondrial genes also exhibited high mutation frequency and relatively high occurrence, consistent with the accumulation of mitochondrial mutations in tumor cells [22]. We further compared the mutated genes with those detected in the BRCA and COAD cohorts from The Cancer Genome Atlas (TCGA) database and found that most mutated genes were shared in the TCGA database (Fig. 2D). However, we observed that tumor driver genes such as *GATA3* and *ESR* in DCIS [23, 24] were almost uniformly distributed spatially, but their corresponding SNVs were only present in the tumor. The SNVs and gene expression exhibited different patterns (Fig. 2E). Chi-square tests of SNVs and RNA with tumor incidence indicated that the occurrence of SNVs is more closely associated with tumor regions (Fig. 2F). Given the close association of SNVs and genes with tumorigenesis, we further examined the distribution of other driver genes and SNVs to provide additional support for this perspective. Despite these driver genes being widely expressed across different spatial regions, the SNVs associated with these driver genes were confined solely to the tumor region (Fig. 2G). Moreover, we

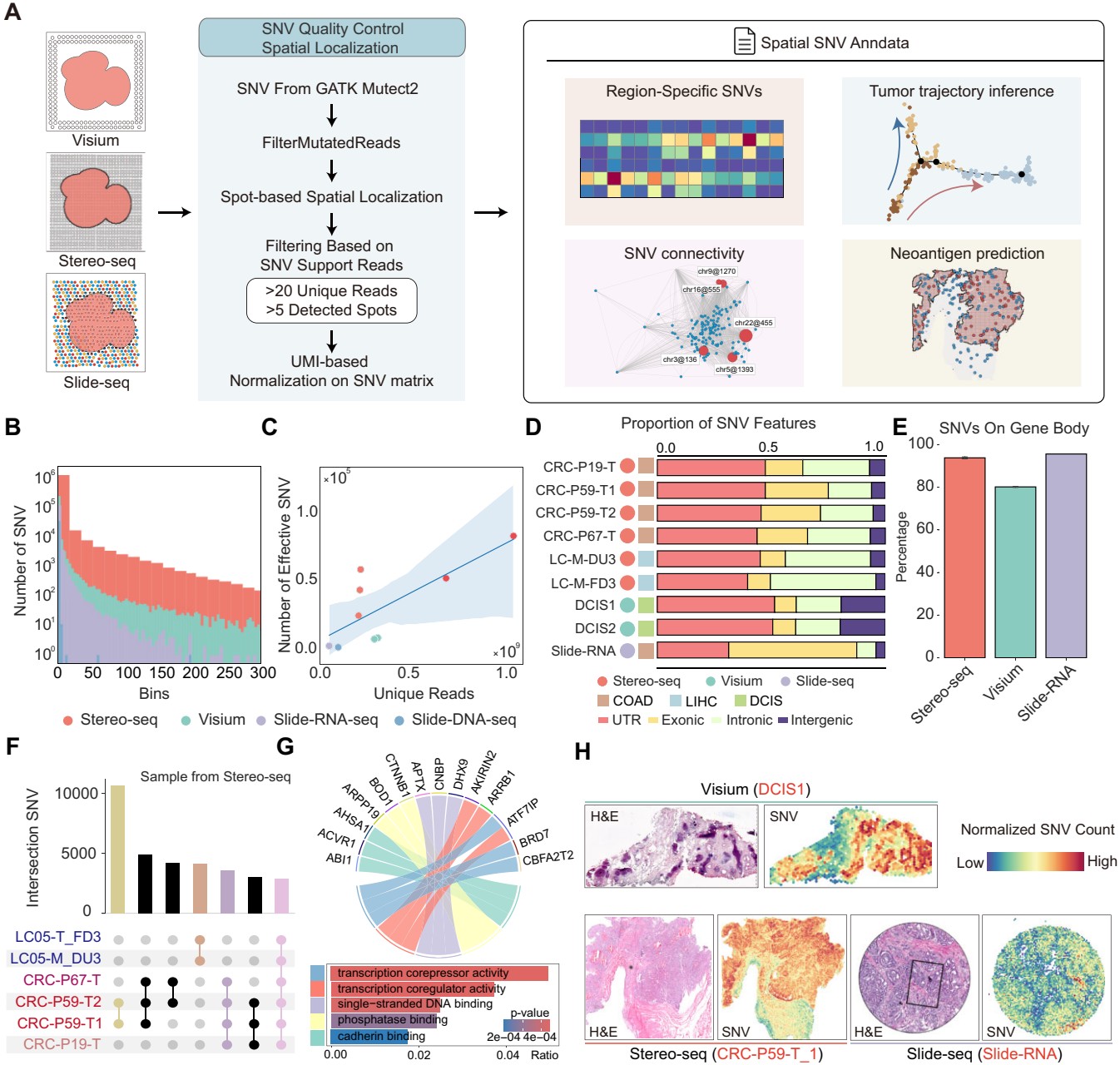

**Figure 1:** SpatialSNV exhibits spatial distribution mutations by calling SNVs from spatial transcriptomics. (A) Schematic depicting the methodology for detecting SNVs from spatial transcriptomics data. (B) Frequency histogram illustrating the distribution of shared SNVs among spatial spots, with representative sections selected for each platform: CRC-P19-T (Stereo-seq), DCIS1 (Visium), and primary human colorectal cancer (Slide-RNA-seq and Slide-DNA-seq). (C) Scatterplot showing the correlation between unique reads (excluding duplicates) from spatial transcriptomics and the number of detected effective SNVs. (D) Stacked percentage bar chart showing genomic regions containing effective SNVs. Circular markers denote the platforms of the samples, and square markers specify the cancer types. (E) Bar plot detailing the proportion of effective SNVs within gene body regions. (F) UpSet plots demonstrating the intersections of effective SNVs among different sections. Yellow bar represents samples from the same patient, while brown and purple bars correspond to samples from liver cancer or colorectal cancer, respectively. Pink bars indicate SNVs that are common in all patients. Sample IDs are colored according to the same patient. (G) Bar plots illustrating GO Term enrichment for Molecular Function associated with effective SNVs, as detailed in panel D. The chord diagram illustrates the genes involved in key GO pathways, highlighting their connections and functional relevance. (H) Hematoxylin and eosin (H&E) staining (left) and spatial visualization (right) of normalized SNV counts across representative samples from various platforms.

performed Pearson correlation analysis between normalized SNV counts and hallmark gene sets from MSigDB [25] on tumor samples. We found that the mutation burden in both DCIS and CRC samples was highly correlated to pathways such as MYC regulation, oxidative phosphorylation, and *KRAS* signaling, which are known to play critical roles in tumorigenesis [26–28] (Fig. 2H). These findings demonstrate that spatially resolved SNVs identified by SpatialSNV can offer genetic information from an additional dimension, extending beyond the insights provided by spatial transcriptomics data.

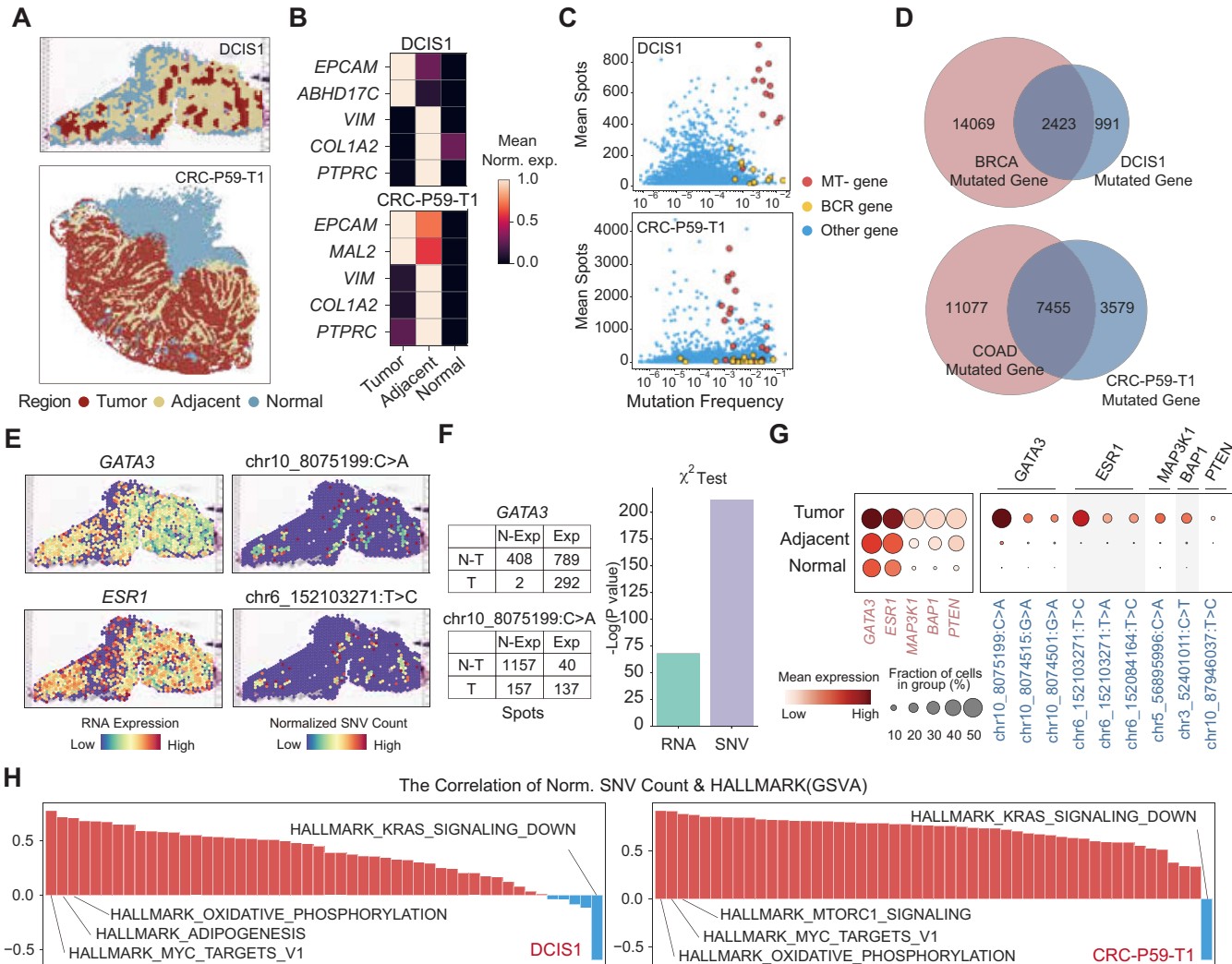

**Figure 2:** Tumorgenesis-associated SNVs inferred from spatial transcriptomics data. (A) Spatial visualization of the region clustered by spatial transcriptomics. (B) Heatmaps of the primary markers of the cluster region as shown in panel A, with expression counts normalized by z-score. (C) Scatterplots demonstrating the mutation frequency of genes alongside the average occurrence of corresponding SNVs across spatial spots. (D) Venn diagrams showing the overlap of mutated genes from spatial transcriptomics with those identified in related cancer types in TCGA. (E) Spatial visualization representing the distribution of driver genes *GATA3* and *ESR1*, along with their corresponding representative SNVs. (F) Bar plot illustrating the correlation between *GATA3* expression and its representative SNVs with the prevalence of tumor regions, with statistical significance assessed using the chi-square test. (G) Dot plot highlighting the expression levels of driver genes *GATA3* and *ESR1* and their corresponding SNVs in different regions of DCIS1. (H) Bar plots demonstrating correlations between various pathways scores and normalized SNV counts per spot. Red bars indicate positive correlations, while blue bars denote negative correlations.

## SpatialSNV helps to trace tumor development *in situ*

SNVs can serve as imprints to trace tumor development. We then explored a DCIS sample previously shown to have 3 subclones [19]. Consistent with the original study, we observed a pronounced clonal distribution from the transcriptome data (Fig. 3A, Supplementary Fig. S5A, B). We also analyzed the CNVs using the inferCNV algorithm, which confirmed the clonal diversity of the 3 tumor subclones (Fig. 3B). Additionally, to determine whether SNVs could reproduce the clustering patterns, we performed dimensionality reduction and clustering on the spots using SNV windows as features. This analysis revealed clustering results similar to those observed with CNVs (Supplementary Fig. S5C). We used SpatialSNV to perform differential analysis on SNVs and observed different SNV profiles of 3 tumor subclones (Fig. 3C). For example, *THOP1* was evenly expressed in 3 subclones, but the associated SNV chr19_2813592:G>A occurred more frequently in

subclone 1. Similarly, *PRSS23* expressed high in subclones 0 and 1, while the corresponding SNV chr11_86808877:T>C occurred predominantly in subclone 1 (Fig. 3D, E). The inconsistency of the spatial patterns between SNVs and gene expression indicated a dynamic clonal development progress within the tumor tissue.

To trace the clonal development *in situ* of tumor cells, we employed the Monocle2 algorithm [29] to construct the developmental trajectory using the differentially distributed SNVs. We found that subclones 1 and 2 might have differentiated from subclone 0 (Fig. 3F), which is consistent with the CNV profiles of the subclones. Subclone 2 appeared to have CNV gains on chr3 and CNV loss on chr6 and chr11, and subclone 1 suffered CNV gain on chr5 (Fig. 3B). Therefore, subclones 1 and 2 would accumulate more mutations during clonal differentiation theoretically. As anticipated, we identified more SNVs in subclones 1 and 2 (Fig. 3G, H). We also observed that the differential SNVs, such as chr7_98385752:A>G and chr19_2813592:G>A, were elevated along

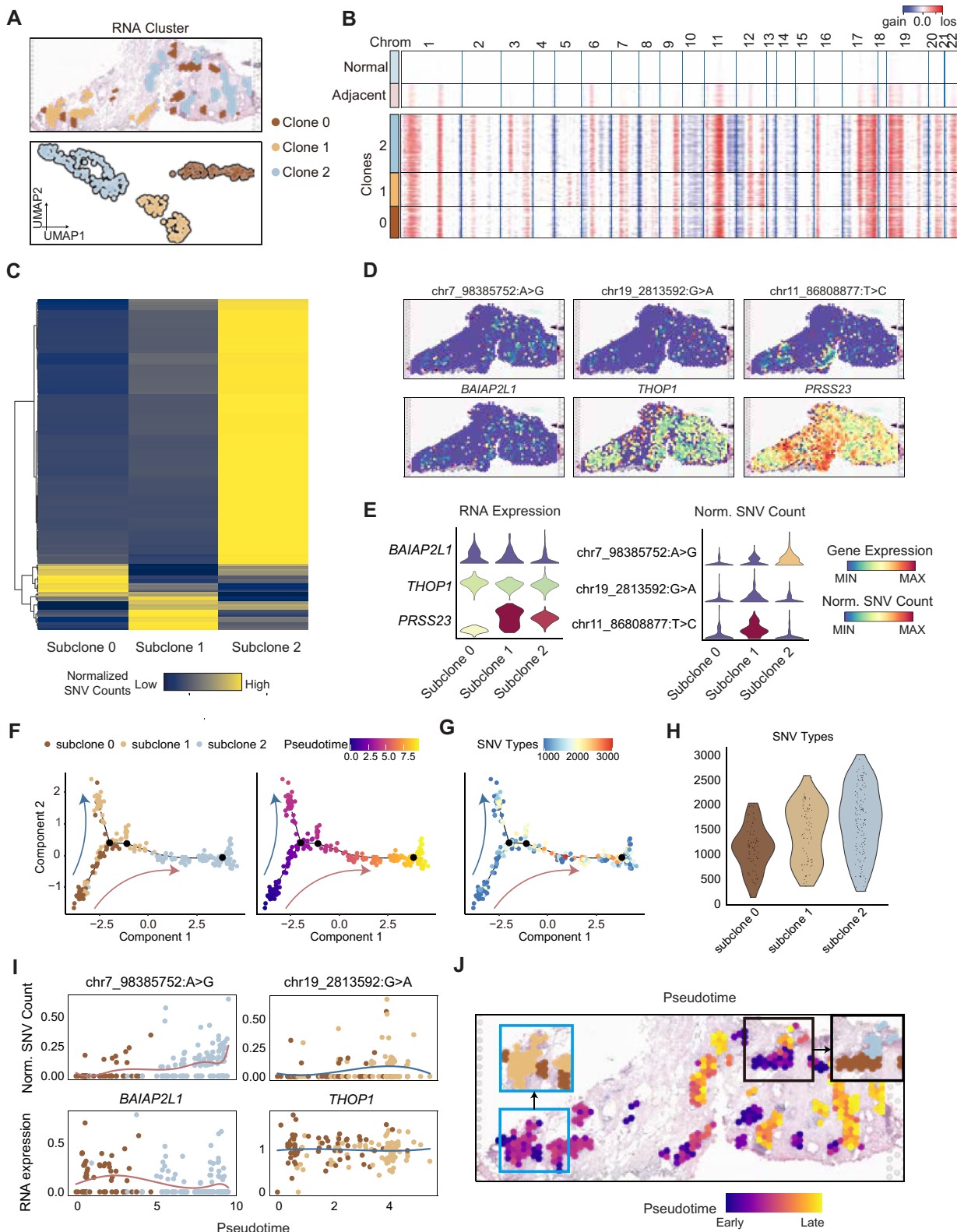

**Figure 3:** SpatialSNV traces the mutation history of different tumor subclones. (A) Top: Spatial visualization illustrating the distribution of the tumor subclones. Bottom: UMAP projection of the tumor subclones. (B) Heatmap generated by inferCNV displaying inferred CNA profiles for 3 tumor subclones. (C) Heatmap showing the normalized SNV count of specific SNVs within 3 tumor subclones at the pseudo-bulk level. (D) Spatial visualization of normalized SNV counts and gene expression. (E) Violin plot showing the expression levels of specific SNVs and genes within the tumor subclones. (F) Spatial spot trajectory generated by SpatialSNV analysis, depicting the pseudotime and evolutionary trajectories of 3 tumor subclones. (G) Spatial spot trajectory showing the accumulation of SNV types along the evolutionary trajectories. (H) Violin plot displaying the number of SNV types across all spots within 3 tumor subclones. (I) Scatterplot displaying the expression variations of tumor-specific SNVs and corresponding genes across different evolutionary branches. (J) Spatial visualization of the pseudotime of tumor subclones.

clonal differentiation trajectories (Fig. 3I), indicating their contributions to the tumor clonal development. Moreover, we mapped the predicted pseudotime to the tumor section and found that tumor subclone 0 gradually differentiated into subclones 1 and 2 in the co-occurring tumor regions (Fig. 3J). Additionally, in subclone 2–specific tumor regions, spots on the tumor margins appeared at a later developmental stage (Fig. 3J), suggesting that subclone 2 was still undergoing rapid evolution. SpatialSNV facilitates the analysis of regional tumor evolution traces imprinted on spatially resolved SNVs.

## SpatialSNV reveals mutational dynamics at the tumor margins

As tumors invade, the tumor margins often exhibit a chronic inflammatory microenvironment [30], which may trigger more mutation events. Consistently, we investigated the genetic landscape of the tumor tissues and observed that tumor margins gained significantly more SNVs than distant normal regions (Fig. 4A, B, Supplementary Fig. S6A). Moreover, we found that the tumor margins had unique mutational profiles (Fig. 4C, D). For example, SNVs chr20_4024544:C>T and chr20_4024581:C>T occurred more frequently in the tumor margins of the CRC-P59-T1 sample and exhibited a clear spatial preference for the tumor margins (Fig. 4D, E). When surveying their genomic loci, we found significantly more sequencing reads covering this region in the tumor margins than in tumor and normal tissues, implying that these SNVs may alter gene expression (Fig. 4F). For instance, the mutation chr5_151661589:G>C is located in the UTR region of *SPARC*, which is highly expressed in metastatic tumors [31], potentially indicating the invasiveness of the tumor margin. Similarly, the UTR mutation chr7_23274960:A>T in *GPNMB* may also influence *GPNMB* expression, thereby promoting tumor metastasis [32] (Supplementary Fig. S6B). We further investigated the genes associated with SNVs predominantly present in tumor margins and found them to be highly related to the extracellular matrix and cell adhesion processes (Fig. 4G), suggesting that the tumor cell in the adjacent margins may help reshape the microenvironment, potentially contributing to tumor immune escape [33]. To verify this, we examined the inflammatory response and hypoxia processes, which are reported to be essential in tumor immune escape [34, 35]. We found that the expression of genes related to both pathways declined along with the normalized SNV count in the tumor margins as the distance increased from the tumor boundary (Fig. 4H, Supplementary Fig. S6C). This suggests that tumor expansion may lead to the emergence of new tumor cells at the margin, which could subsequently give rise to novel SNVs.

The relationship between the characteristics of the tumor margin microenvironment and tumor progression remains largely unknown. To address this, we plotted the normalized SNV counts against the distance to the tumor boundaries. We found that SNV accumulation was negatively correlated with the distance from the tumor boundary (Fig. 4I), a phenomenon similar to the higher somatic copy number alteration (SCNA) burden observed in the tumor center [36]. Furthermore, we observed varying rates of decline in the mutational dynamics at the tumor margins across tumor samples and found that the rate of decline in mutational dynamics is highly correlated with the immune activity of the tumor section (Fig. 4J). Tumors with higher immune activity exhibited a slower decline in mutational dynamics, suggesting that the tumor margin may harbor more neoantigens, leading to the accumulation of immune cells.

## Spatially correlated SNVs contribute to reshaping tumor microenvironments synergistically

Subsequently, we aimed to identify specific SNVs responsible for reshaping the microenvironment of the tumor margins. To reduce the sparsity of the spatial SNV matrices, SpatialSNV partitioned the genome into 100,000-bp windows and aggregated SNV events in each window for each spatial spot. We hypothesized that SNVs occurring in close spatial proximity are genetically associated. Therefore, we calculated the spatial correlations between SNV windows and constructed a connectivity graph of the SNV windows for SNV cluster detection (Fig. 5A). Using the Leiden algorithm, we partitioned SNVs into different groups, termed SNV groups. SNV groups exhibited more significant spatial distribution patterns than the individual SNVs they contained (Fig. 5B, Supplementary Fig. S7A). The spatial patterns of SNV groups differed from each other (Fig. 5C), indicative of different biological processes. Consequently, we examined the corresponding genes of each SNV group and found them to be involved in distinct GO terms. For example, SNV group 1 was primarily associated with chemokine activity, while SNV group 2 mainly affected components related to the extracellular matrix (Fig. 5D), consistent with the specific spatial distribution of SNV group 2 at the tumor margins. Considering that the extracellular matrix is highly related to epithelial–mesenchymal transition (EMT) [37], we speculated that SNVs within SNV group 2 may contribute to the tumor EMT process. We then performed gene set variation analysis (GSVA) [38] analysis and confirmed that genes associated with SNVs within SNV group 2 were highly enriched in the EMT pathway (Fig. 5E). Similar SNV groups were also observed across tumor samples. For instance, SNV group 1 of the CRC-P19-T sample was distributed at the tumor margins and was also highly enriched in the EMT pathway (Supplementary Fig. S7B–D). These data suggested that genes involved in similar biological processes may mutate synergistically.

To reveal the potential connections between SNV groups, we analyzed the levels of calculated connectivity between SNVs (Fig. 5F). We found that some SNV windows had higher connectivity than others. In SNV group 2, for example, chr22@328, chr7@232, and chr19@449 were associated with TIMP3, GPNMB, and APOE, respectively, which are marker genes of tumor-associated macrophages (TAMs) (Fig. 5G). This suggests that SNV group 2 may reflect the presence of TAMs in the tumor microenvironment and the genomic alterations driven by their strong transcriptional activity. Therefore, we examined the TAM signature and observed that the TAM signal was highly colocalized with SNV group 2 (Fig. 5H). Moreover, we found that the pleiotrophin (PTN) signaling pathway was also highly expressed around these regions, suggesting that SNV group 2 may be responsible for the TAM microenvironment at the tumor margins. In the CRC-P19-T section, SNV group 1 contained highly connected SNV windows such as chr22@455, chr3@136, and chr16@555 (Supplementary Fig. S7E), as well as associated with tumor metastasis–related genes of *FBLN1*, *FBLN2*, and *MMP2* [39–41], respectively. Another highly connected SNV window, chr5@1393, is associated with *MZB1* (Supplementary Fig. S7F), an endoplasmic reticulum stress-related gene in B cells [42]. We also observed B-cell and plasma cell marker genes such as *JCHAIN* and *CD27* in SNV group 1 enriched region (Supplementary Fig. S7G), suggesting high transcriptional activity of B cells and plasma cells. These findings showed that SpatialSNV is able to identify SNV groups composed of SNVs with similar spatial patterns and biological functions, which contribute to the tumor microenvironment synergistically.

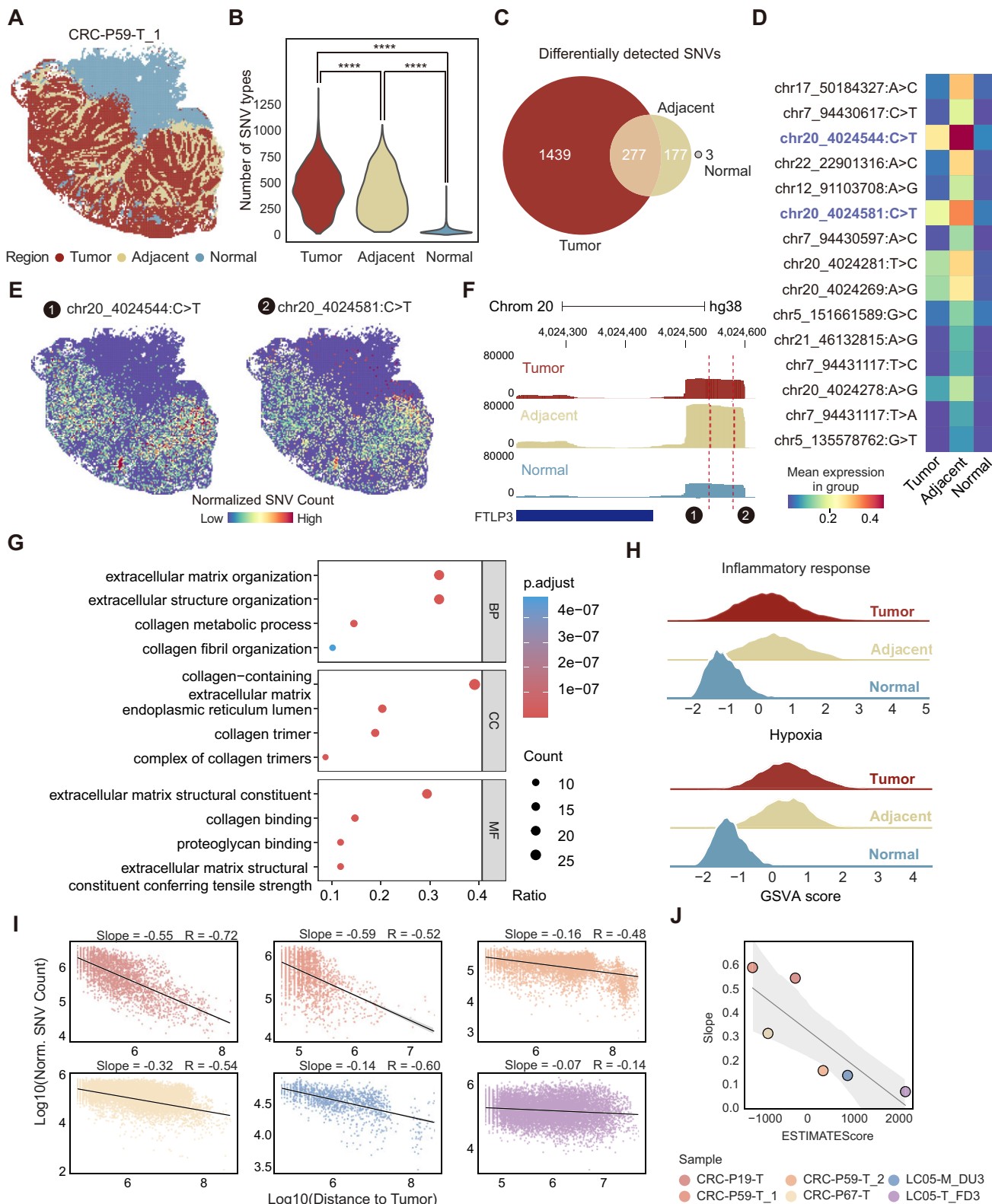

**Figure 4:** The immune microenvironment shapes mutational pressure in tumor margins. (A) Spatial visualization of the regions in the Stereo-seq section CRC-P59-T1. (B) Violin plots showing the distribution of SNV types across different regions with statistical significance determined by *t*-tests; ****$P < 0.01$. (C) Venn diagram illustrating the differences in nonsynonymous SNVs between distinct regions. (D) Heatmap displaying the normalized counts of the top 20 spatial SNVs specific to adjacent margins. SNVs located in intergenic regions are highlighted in blue. (E) Spatial visualization showing specific intergenic SNVs from panel D. (F) Reads track visualizing the alignment of reads in intergenic regions across different regions. (G) GO term analysis of genes affected by intergenic SNVs, categorizing into Biological Process (BP), Cellular Component (CC), and Molecular Function (MF). (H) Ridgeline plot illustrating specific pathway scores calculated through GSVA across different regions. (I) Scatterplot showing the relationship between normalized SNV count and tumor distance in each spatial spot, with both axes log-transformed. $R^2$ represents the Pearson correlation coefficient. (J) Scatterplot displaying the correlation between the slopes of normalized SNV counts versus tumor distance from panel I and estimate scores. Translucent bands around the regression line indicate the confidence interval for the regression estimate.

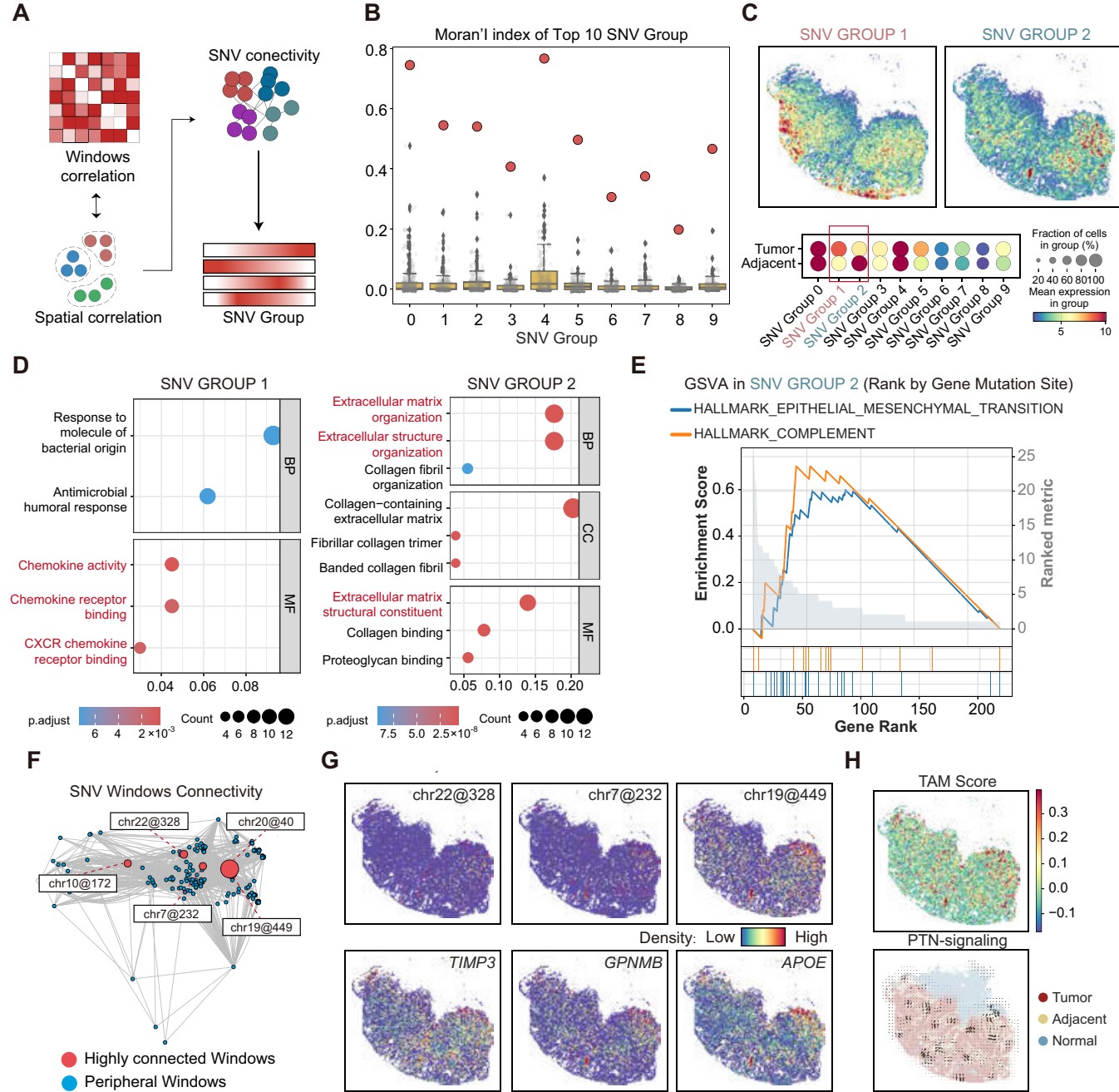

**Figure 5:** Spatially correlated SNVs contribute to shaping tumor microenvironments synergistically. (A) Schematic depicting the methodology for detecting groups of SNVs. (B) Boxplot of Moran's index for the top 10 SNV groups. Gray dots represent Moran's index of individual SNVs within each group, while red dots indicate Moran's index for each SNV group. (C) Top: Spatial visualization of SNV group 1 and SNV group 2. Bottom: Dot plot of the top 10 SNV group expression in tumor and adjacent margins. (D) GO term analysis of genes within the SNV windows of SNV group 1 and SNV group 2. (E) Gene set variation analysis of the enriched pathways for genes within SNV windows of SNV group 2. (F) Network graph illustrating the connectivity among SNV windows in SNV group 2, with points representing SNV windows and line lengths indicating the degree of correlation. (G) Spatial visualization of the spatial distribution of highly connected SNV windows (top) and the representative genes contained within these windows (bottom). (H) Spatial visualization of the TAM score (up) and the trend of the PTN signaling pathway across the spatial domain (bottom).

## SpatialSNV identifies tumor region–specific exonic nonsynonymous SNVs as potential neoantigens

Tumorgenesis-associated SNVs may serve as therapeutic targets. Specifically, those exonic nonsynonymous SNVs may result in neoantigens and subsequently activate the adaptive immune system. To identify potential neoantigens from exonic nonsynonymous SNVs *in situ*, SpatialSNV used sliding window translation

of exonic mutations to predict potential neoantigen peptides and filtered out normally expressed peptides by searching public databases and protein from the human coding DNA sequence (CDS) (Fig. 6A). In CRC samples (CRC-P59-T1, CRC-P59-T1, CRC-P67-T), we found thousands of mutations occurring in gene exon regions, most of which were nonsynonymous SNVs (Fig. 6B). We observed that these nonsynonymous SNVs from different samples of the same patient overlapped significantly, while those from other patients also shared a substantial proportion of nonsynony-

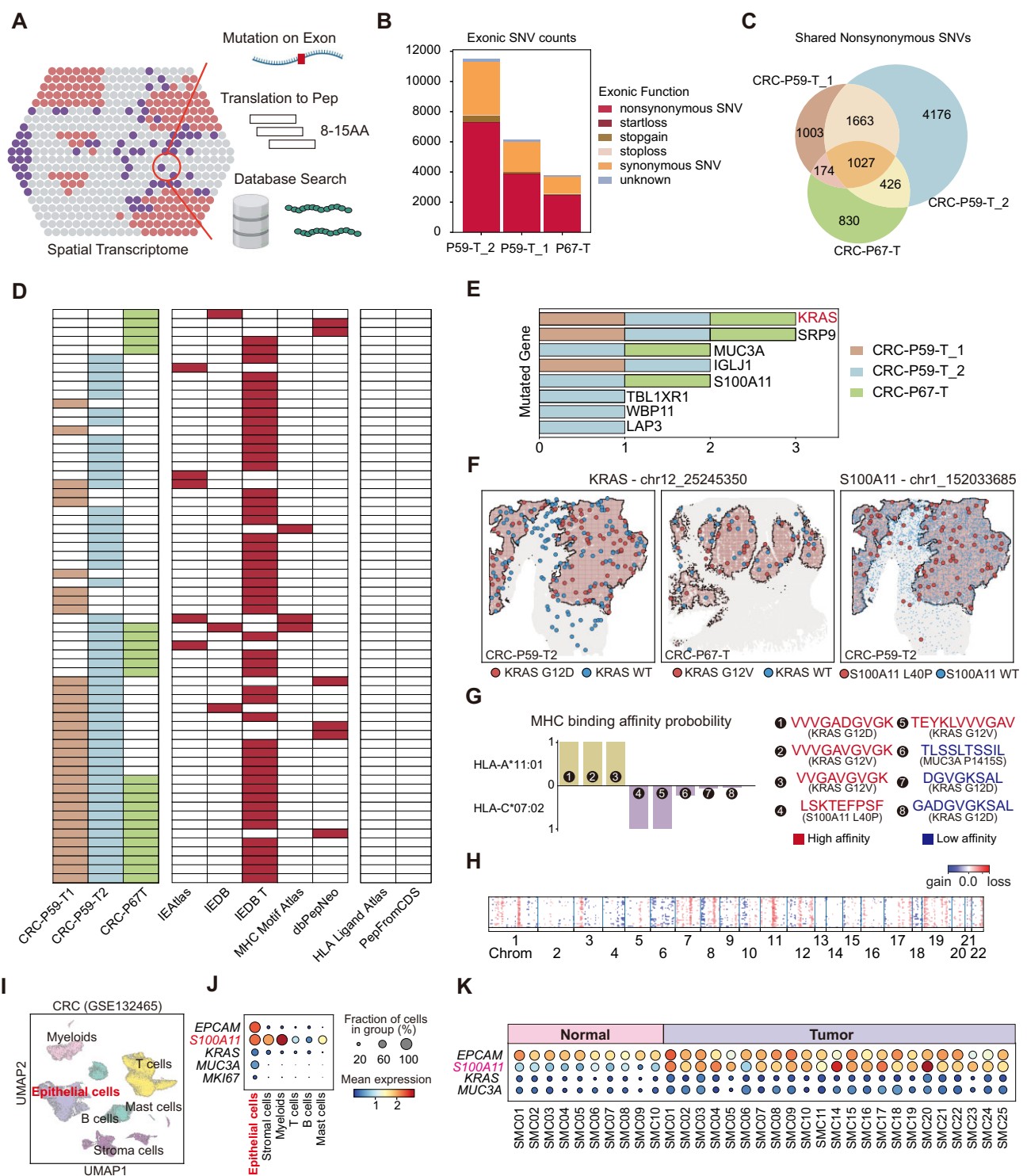

**Figure 6:** SpatialSNV reveals potential neoantigens in tumors. (A) Schematic illustrating the methodology for constructing mutated peptides. (B) Stacked bar chart showing the number of effective SNVs occurring on exons in colorectal cancer samples, with layers representing different patterns of impact on protein coding caused by SNVs. (C) Venn diagram depicting the quantity and shared extent of nonsynonymous SNVs across different samples. (D) Heatmap displaying mutated peptides that are supported by the common epitope database and their sharing levels among samples. (E) Bar plot showing the detection of genes originating the mutated peptides featured in panel D across different samples. (F) Spatial visualization of wild-type and mutated *KRAS* or *S100A11* spots. (G) Bar plot illustrating the predicted binding affinity of mutated peptides to different HLA types. (H) Heatmap generated by inferCNV depicting the inferred copy number alteration (CNA) profiles for spots exhibiting the chr1_152033685 (S100A11[L40P]) mutation. (I) UMAP projection of the single-cell data from GSE132465. (J) Dot plot of the expression of primary markers for tumor and *S100A11* across different clusters. (K) Dot plot of the expression of primary markers for tumor and *S100A11* across different samples.

mous SNVs (Fig. 6C), indicating potential therapeutic neoantigens for CRC.

To identify potential therapeutic neoantigens from the nonsynonymous SNVs, we used the HLA Ligand Atlas database [43] and protein CDS to exclude epitopes presented in healthy individuals. Subsequently, we searched the IEDB [44], MHC Motif Atlas [45], and dbPepNeo [46] to find known peptides presented by HLAs (Fig. 6D). We identified 64 potential neoantigens from the 3 CRC samples (Fig. 6D, Supplementary Table S1). We then examined the corresponding genes of these neoantigens and found that mutated peptides from *KRAS*, *SRP9*, *MUC3A*, and *S100A11* appeared in multiple samples (Fig. 6E). *KRAS* mutations appeared in approximately 40% of patients with CRC, predominantly KRAS$^{G12D}$ and KRAS$^{G12V}$ mutations [47]. T cells targeting KRAS$^{G12D}$ and KRAS$^{G12V}$ have been proven to control tumors efficiently [48, 49]. *MUC3A* can promote the progression of colorectal cancer through the PI3K/Akt/mTOR pathway [50] and has been predicted to be a potential chimeric antigen receptor (CAR) T-cell therapy target [51]. We traced the spatial distribution of these nonsynonymous mutations and observed that although *KRAS* and *S100A11* were expressed in various regions, the mutated gene form was mainly restricted in the tumor regions (Fig. 6F, Supplementary Fig. S8A), suggesting potential neoantigens.

Mutated peptides should be presented by the HLA molecules to serve as neoantigens. We determined the HLA types of the patients from the spatial transcriptomics data using T1K [52] and found both patients to be HLA-A∗11:01 and HLA-C∗07:02 allele types (Supplementary Table S2). We then performed HLA affinity prediction of the mutated peptides using TransPHLA [53]. We found that mutated KRAS$^{G12D}$ and KRAS$^{G12V}$ peptides were predicted to have strong affinity with HLA-A∗11:01, with a KRAS$^{G12V}$ peptide also predicted to be presented by HLA-C∗07:02 (Fig. 6G). Considering that HLA-A∗11:01 is one of the most abundant HLA alleles, especially in East Asia [54], these mutated KRAS peptides can potentially be good T cell receptor (TCR) T-cell therapy targets. Besides mutated KRAS peptides, we also found a peptide "LSK-TEFPSF" derived from S100A11$^{L40P}$ to have a strong affinity with HLA-C∗07:02 (Fig. 6G). To verify whether *S100A11* is expressed in CRC tumor cells, we explored the CNV profile of the spots with S100A11$^{L40P}$ mutation and found them to have significant genome alterations (Fig. 6H, Supplementary Fig. S8B). Moreover, we examined the *S100A11* expression in published CRC single-cell sequencing data [55] and confirmed that *S100A11* is highly expressed in CRC tumor cells (Fig. 6I–K, Supplementary Fig. S8C). These data suggested that S100A11$^{L40P}$ can serve as a potential therapeutic target for precision medicine design.

## Discussion

In this study, we developed SpatialSNV for calling and analyzing effective SNVs from spatial transcriptomics data. Incorporating spatial mutation information provides a more comprehensive tissue context of tumors. We observed significant platform bias when calling SNVs from spatial transcriptomics data generated by different techniques, likely attributed to the varying RNA capture efficiency and sequencing depth among the platforms. Notably, Stereo-seq data tend to have higher total UMI counts, enhancing effective SNV calling [56]. Although SpatialSNV was developed on spatial transcriptomics data, calling SNVs from transcriptomics data still has limitations: a relatively higher false-positive rate [6, 57] and high correlation with gene expression, compared to DNA-seq data. We selected SNVs supported by multiple unique reads and observed across several spots to reduce the false-positive er-

ror rate. Furthermore, we normalized the SNV counts against the sequencing depth (total UMI count) at each spot, which has been proved to reduce the influence of gene expression on SNV quantities efficiently. Thus, our study provides a relatively robust method to recover SNV information from spatial transcriptomics data. Moreover, SpatialSNV can also be modified for spatial DNA sequencing data such as Slide-DNA-seq. Due to the poor sequencing depth, we could not recover enough SNVs for subsequent analysis, demonstrating that calling SNVs from spatial DNA-seq data remains a big challenge. More precise spatial SNV analyzing will need a further developed spatial DNA-seq technology with higher data quality and deeper sequencing depth. It is also important to note that platforms such as Visium, Stereo-seq, and Slide-seq are poly(T) capturing-based sequencing methods, which may introduce a strong bias toward SNVs near the 3' end. Recent spatial techniques, such as Patho-DBiT [58], could provide a relatively even gene body coverage. This experimental advancement may help to improve the efficiency in capturing variations across the gene body.

Many cancer cells at solid tumor margins exhibit reversible invasiveness, coordinated by a developmental regulatory program known as EMT [59]. Specific conditions, such as hypoxia and factors such as cytokines secreted by stromal cells, can induce EMT, thereby promoting invasiveness [27]. EMT regions are associated with tumor progression and often coexist with immune responses. A high degree of immune infiltration is closely linked to EMT, suggesting that EMT progress closely interacts with the immune system. Our data showed that the mutational dynamics of the tumor margins are involved in hypoxic and immune regulatory processes, which are essential in shaping the tumor microenvironment and promoting tumor expansion [60]. We recognized that mutations at the tumor margins may originate from newly proliferating tumor cells, which may influence the tumor microenvironment in the peripheral regions. However, it is important to note that somatic mutation frequencies are also high in normal nonblood tissues [61]. Therefore, when examining the mutational landscape of the tumor microenvironment, the extent of the influence of tumor cells on this landscape remains unclear. Although the tumor center in clear cell renal cell carcinoma (ccRCC) has been characterized by proliferation, necrosis, and hypoxia [36], the tumor margin exhibits significant collagen deposition, leading to increased ECM density and stiffness, which further reduces oxygen supply and creates a hypoxic environment [62]. By integrating SNVs with the spatial context, we identified spatially correlated SNVs, termed SNV groups. We found that SNVs within the same group tend to be involved in similar biological processes. Moreover, we noticed that specific SNV groups are associated for the EMT process of the tumor margin regions, which are conserved across different tumor samples, indicating that these SNV groups may be critical factors in regulating the tumor margin environment. We observed these intriguing phenomena in the collected samples, but the potential role of SNV groups in EMT requires more data to support, as does the role of SNVs at the tumor margin. Moreover, the biological significance of SNV groups may require further investigation. We hypothesize that SNVs appearing in other cells may indicate high transcriptional activity in these cells within the tumor microenvironment, leading to certain genetic mutations. Since SNVs are essential sources of tumor-specific neoantigens, these SNVs can potentially be used as therapeutic biomarkers targeting tumor margin regions.

Although SNVs are the most prevalent type of mutation at the genomic level in tumor cells [63], CNVs involving tumor driver genes are generally regarded as the characteristics of many can-

cers [64, 65]. Analyzing SNVs and CNVs at the single-cell level allows for the reconstruction of tumor evolutionary pathways [66, 67], aiding in identifying tumor driver genes. Notably, tumor-specific neoantigens, resulting from mutations in tumor cells, present promising targets for cancer immunotherapy [63]. Although neoantigens can arise from structural variations such as CNVs [68], most tumor neoantigens are derived from SNVs [69, 70]. In our study, we used SpatialSNV to identify tumor region–specific SNVs in CRC samples. By focusing on the nonsynonymous SNVs in exonic regions of protein-coding genes, we found potential neoantigens of the tumors and predicted their HLA affinities. With this approach, we identified KRAS$^{G12D}$, KRAS$^{G12V}$, and S100A11$^{L40P}$ mutations as potential therapeutic targets. KRAS$^{G12D}$ and KRAS$^{G12V}$ are common mutations used as targets for universal therapy strategies, while S100A11$^{L40P}$ is more likely an individually expressed mutation, which can be targeted in personalized therapies. We provided a neoantigen prediction guidance in this study, but did not perform experimental validations. However, the *KRAS* associated neoantigens, which have been widely documented in the literature to be genuinely present in colorectal cancer, could demonstrate the reliability of our methodology to some extent. Therefore, spatially resolved SNVs are useful in finding therapeutic targets for precision medicine design.

## Methods

### Spatial transcriptomics data processing

For Visium (RRID:SCR_023571) spatial transcriptomics, we employed Space Ranger version 1.3.1 (RRID:SCR_025848) to generate binary alignment map (BAM) files, utilizing the hg38 reference genome (GRCh38-2020-A provided by 10x Genomics). For the analysis of Stereo-seq data, we utilized the Stereo-seq Analysis Workflow [71] (SAW V4.0, RRID:SCR_025001), with default parameters to align sequences against the hg38 reference genome, using the GRCh38.p12 GTF file for gene annotations. Slide-DNA-seq was performed in accordance with the established protocols. For Slide-RNA-seq, we implemented the Drop-seq tools integrated within the Slide-seq Tools suite, aligning to the hg38 reference genome, consistent with the reference used for Visium analyses.

Analysis and visualization of spatial transcriptomics data were conducted using Scanpy [72] (RRID:SCR_018139). For the Stereo-seq platform, gene counts were aggregated within $100 \times 100$ spot blocks to construct the bin100 gene count matrix. Spots containing fewer than 100–200 genes and genes present in fewer than 10 spots were excluded to ensure data quality. Additional spot filtering was applied based on metrics such as n_genes_by_counts and pct_counts_mt, which were tailored to the specific characteristics of each sample. Counts were normalized using the normalize_total function in Scanpy, followed by dimensionality reduction through principal component analysis (PCA). The top 15 principal components were selected to construct neighbor graphs with a parameter setting of 15 neighbors. Clustering of spatial spots was performed using the Leiden algorithm to identify potential clusters. Spatial domains within the tissue, such as tumor region, adjacent margins, and normal region, were preliminarily categorized using marker genes, including *MKI67*, *TP53*, *EPCAM*, *VIM*, and *PTPRC*.

### Inferred copy number variation (inferCNV) analysis

To validate the effectiveness of tumor regions identified from spatial transcriptomics, we used inferCNV to verify copy number variations in different areas further to determine the accuracy of the region segmentation. The gene matrix was used as input for inferCNV (inferCNV of the Trinity CTAT Project, RRID:SCR_025804). Simultaneously, chromosomal positions for all genes are annotated by searching the GTF file, which also serves as input for inferCNV. The cnv.tl.infercnv function was set with the following parameters: lfc_clip = 3, window_size = 250, and exclude_chromosomes=("chrX,""chrY"). The normal region was selected as a reference for inferCNV analysis.

### Expression track visualization

The BigWig files are created from BAMtype files using the program bamCoverage from deepTools (RRID:SCR_016366) [73] with parameters "–binSize 1," "–normalizeUsing RPKM," " –exactScaling," and "–minMappingQuality 10." All BigWig data were stored in CyVerse (RRID:SCR_014531) [74] for uploading to the UCSC genome browser (RRID:SCR_005780) [75] for visualization.

### Somatic mutation calling with mutect2

Aligned SAM files were converted to BAM files and sorted by coordinate using Samtools (RRID:SCR_002105) [76] (version 1.11). To remove duplicated reads on spatial transcriptomics data, we utilized a custom Python script to combine UMI and barcode tags. The MarkDuplicate function of Picard (RRID:SCR_006525) was applied to remove duplicated reads from each BAM file.

GATK's Mutect2 [16] (RRID:SCR_026692, version 4.2.6.1) was used to call mutations according to their somatic pipeline. We first used addOrReplaceReadGroups to add group information to the BAM file, which facilitates the recognition of tumor files by Mutect2. Following GATK's recommendations for calling mutations on transcriptomes, we proceeded with splitNCigarReads and ApplyBQSR to process the BAM files, where ApplyBQSR's –knownsites parameter used dbsnp_151.hg38.vcf.gz from the GATK resource bundle. Mutect2's tumor-only mode was employed for mutation calling in spatial transcriptomics with default parameters, where the germline resource and Panel of Normals were af-onlygnomad.hg38.vcf.gz and 1000g_pon.hg38.vcf.gz respectively, from the GATK resource bundle. Although Mutect2 is not the optimal choice for calling mutations from DNA, we utilized Mutect2 without the splitNCigarReads option for SNV calling in Slide-DNA-seq.

Following SNV calling, we employed FilterMutectCalls to filter the SNVs, excluding those annotated with tags such as weak_evidence, germline, strand_bias, slippage, contamination, and panel_of_normals in the VCF file. We then traced the reads aligned to each SNV locus, extracting the coordinates of reads that contained the alternative base. These coordinates were used to construct a binary matrix, indicating the occurrence of specific SNV events at each spot. Specifically, on the Stereo-seq platform, SNV counts were aggregated within blocks of $100 \times 100$ spots to form the bin100 SNV count matrix. For an SNV to be considered effective, it was required to be supported by at least 20 unique reads and appear in at least 5 spots. Effective SNVs were annotated using ANNOVAR (RRID:SCR_012821) [77].

### Gene mutation frequency

We quantified the number of effective SNV sites per gene to assess gene mutation frequency and normalized these figures by gene length. A higher gene mutation frequency suggests an increased mutational pressure on that gene.

## Normalization of SNV counts

For the calculation of the normalized SNV cont, in order to eliminate the impact of sequencing depth on SNV counts, we used the UMI counts of each spot to normalize the expression of SNVs.

$$\text{Norm SNV}_i = \log\left(\frac{s_i}{\sum_{j=1}^{n} u_j} + 1\right)$$

where $i$ is the $i$th SNV, $s_i$ represents the SNV count number, $u_j$ represents the UMI count number, and $n$ is the total number of genes.

## Identification of tumor subclones and pseudotime analysis

The expression matrix from spots designated as tumor regions was extracted. This matrix was then normalized and subjected to dimensionality reduction clustering via Scanpy, consistent with the methods described in the "Spatial Transcriptomics Data Processing."

Next, normalized SNV count matrices were extracted from spots designated as tumor regions. Analytical objects were created using Monocle2's GaussianFF model (RRID:SCR_016339) [29]. SNVs for inferring spot pseudotime were identified by Scanpy as differential SNVs (log-fold change ≥ 1, $P < 0.5$). Pseudotime ordering of different spots was determined using reduceDimension with norm_method=none and orderCells function. Visualization of SNVs across evolutionary branches was facilitated using Python.

## Correlation between normalized SNV counts and distance

The KDTree function in the Python package scipy was used for nearest-neighbor search (NNS). The average distance and feature expression of the 5 ($k = 5$) nearest tumor region spots around each adjacent margins spot were calculated. We employed the scipy.stats.linregress function to fit a linear model between the logarithm of normalized SNV count and the logarithm of average distances from the tumor for each spot. The correlation between these was quantified using the coefficient of determination ($R^2$), derived from the squared Pearson correlation coefficient provided by the regression.

## Enrichment analysis

For GO analysis and visualization, we utilized the R package clusterProfiler (RRID:SCR_016884) [78]. GO enrichment analysis was conducted using the enrichGO function, targeting specific gene sets with the following key parameters: pAdjustMethod was set to "BH" for multiple testing correction, and qvalueCutoff was established at 0.01 to filter significant terms.

The Gene Matrix Transposed File Format (GMT) file, h.all.v2023.1.Hs.symbols.gmt, was sourced from the MSigDB [79] database to facilitate these analyses. For GSVA analysis and visualization, we employed the Python package gseapy (RRID:SCR_025803) [80]. The gp.prerank function was utilized for GSVA analysis on specified gene sets, with parameters configured as min_size = 5, max_size = 1,000, and permutation_num = 1,000 to ensure robust statistical interpretation.

For GSEA analysis and subsequent visualization, the raw expression matrix from spatial transcriptomics data was first converted into a Seurat [81] object. Normalization was performed using the NormalizeData function. Subsequent GSEA analysis was conducted using the gsva function from the GSVA [38] R package,

with kcdf set to "Gaussian" to assess the enrichment of gene sets in the context of our spatial transcriptomics data.

## Assessment of immune scoring in spatial transcriptomics

To evaluate the proportion of immune cells in each section, we utilized the R package estimate [82]. Initially, gene count matrices for each section were aggregated to create pseudobulk data. These pseudobulk data were normalized using counts per million (CPM) normalization followed by log transformation. Immune cell proportions were then assessed using the estimateScore function, specifying the platform="affymetrix" parameter.

## Calculation of SNV spatial correlations and construction of SNV groups

Due to the abundance of SNV features and the proximity of some SNVs on the genome, we implemented a sliding window approach to aggregate SNVs. SNVs were consolidated into windows of 100,000 bp each. We hypothesized that the co-occurrence of SNVs in space may indicate alterations in certain biological processes. This phenomenon of co-occurrence makes UMI normalization less crucial; therefore, we utilized unique counts of SNVs to represent the expression levels of SNV windows. On the resulting SNV windows counts matrix, distances between spots were computed using the K-nearest neighbors (KNN) method. Additionally, to refine the SNV windows matrix, we applied a Gaussian weighting scheme to attenuate the influence of distant neighbors, enhancing the accuracy and relevance of the spatial genomic data,

$$M_w = W \times M$$
$$W_{ij} = \exp\left(-\frac{D_{ij}^2}{2\alpha^2}\right)$$

where $M$ represents the SNV windows count matrix, and $W$ is a matrix of weights applied to adjust these counts based on spatial proximity between spots. Each weight $W_{ij}$ is calculated using a Gaussian function of the squared distances between spots, moderated by a scaling parameter $\alpha$. $M_w$ represents the adjusted SNV windows matrix.

Due to the high sparsity in the SNV windows matrix, we focused on spots representing the top 50% of expression levels for each window. This approach allowed us to calculate correlations specifically among these more informative spots, thereby optimizing our analysis efficiency,

$$W_{corr} = M_w^T \times M_w$$

where $W_{corr}$ represents the spatial correlation between SNV windows, and $M_w$ represents the transposed matrix of adjusted SNV windows.

After isolating the top 50% of the most correlated windows for each SNV window, we refined the correlation matrix using Gaussian decay. A directed graph was then constructed, with nodes representing the SNV windows and edge weights derived from the adjusted correlations. Then, we applied the Leiden algorithm to cluster SNVs, setting the resolution parameter at approximately 5 to ensure distinct SNV groups were identified. This approach was chosen to preserve groups of SNVs with strong internal correlations. Finally, the UMI count for each spot is used to normalize the SNV group counts to eliminate the effects of sequencing depth.

## Identification of representative genes in SNV windows

Since SNV windows are generated by integrating SNVs, extracting representative genes from SNV windows involves focusing on the gene locations of all SNVs within these windows. We disregard those SNV windows that contain only intergenic SNVs. When all SNVs in a window are located within the same gene, that gene is designated as the representative gene. If an SNV window encompasses multiple genes, we select the gene whose expression pattern aligns with that of the entire SNV window as the representative gene.

## Calculation of Moran's I

Moran's I was calculated using Squidpy's (RRID:SCR_026157) [83] spatial_neighbors and spatial_autocorr functions to assess the spatial autocorrelation of SNV groups and individual SNVs.

## TAM signature

To calculate the spatial signature of TAMs, we selected *CD68*, *CD14*, *MMP2*, *MMP9*, *VTCN1*, *CD163*, and *MRC1* as the gene set for TAMs. The scores were computed using the sc.tl.score_genes function in Scanpy.

## Cell communication in spatial transcriptomics

COMMOT [84] was utilized to analyze and visualize cell communication within spatial transcriptomics data. Initially, the human-specific cell communication database from CellChat was selected using the ligand_receptor_database function. Potential ligand–receptor pairs were filtered using the filter_lr_database function with a minimum cell percentage threshold set at 0.01. Subsequent analysis of spatial cell communication was conducted through the spatial_communication function, applying parameters such as dis_thr=500 for the distance threshold, heteromeric=True to consider heteromeric complexes, and pathway_sum=True to aggregate signals by pathway.

## Neoantigen prediction

We extracted nonsynonymous SNVs annotated by ANNO-VAR to construct a potential mutant peptide database. Utilizing amino acid mutation annotations provided by ANNO-VAR, we extracted 15 amino acids surrounding each mutated site from the CDS reference matching the reference genome (GCF_000001405.38_GRCh38.p12). A sliding window approach generated potential mutant peptides ranging from 8 to 15 amino acids. Using a Python script, we obtained all peptides of 8–15 amino acids in length from the reference CDS and peptides from the HLA Ligand Atlas database [43] to filter epitopes derived from normal proteins. Concurrently, we employed IEDB [44], MHC Motif Atlas [45], and dbPepNeo [46] to identify known peptides presented by HLAs. Additionally, the affinity between neoantigens and HLA is predicted using T1K with default parameters.

## Availability of Source Code and Requirements

Project name: SpatialSNV
 Project homepage: https://github.com/YoungLi88/SpatialSNV
 Operating system(s): Linux
 Programming language: Python
 Other requirements: Python 3.8 or higher
 License: MIT License
 RRID:SCR_026221

## Additional Files

**Supplementary Fig. S1.** The quality control of spatial RNA and SNV data across various sections. (A) Table presenting the data sources for each sample. (B) Bar plot displaying the number of unique reads supporting SNVs for each section. (C) Violin plot illustrating the distribution of basic quality control metrics for spatial transcriptomics data across all sections, including the number of genes by counts (n_gene_by_counts), total counts, and percentage of mitochondrial counts (pct_count_mt). (D) Bar plot showing the quantity of clean reads across different platforms, with error bars indicating variability among sections. (E) Bar plot depicting the number of effective reads for all section.

**Supplementary Fig. S2.** Relationship between spatial transcriptomics and spatial SNVs. (A) Spatial visualization illustrating the total RNA UMI counts (left), SNV counts (mid), and normalized SNV per UMI (Normalized SNV Count) (right). (B) Scatterplots correlating SNV count and normalized SNV count with UMI count per spot, including the Pearson correlation coefficient ($R^2$).

**Supplementary Fig. S3.** The definition of spatial region. (A) Spatial visualization of region clustered by spatial transcriptomics and the dot plot showing the main marker of each cluster region.

**Supplementary Fig. S4.** The inferCNV results of each section. (A) Heatmap generated by inferCNV displaying inferred CNA profiles for each region clustered by spatial transcriptomics.

**Supplementary Fig. S5.** Transcriptional characteristics of tumor subclones in the DCIS1 section. (A) Heatmap of the top 10 differential genes of 3 subclones. (B) Spatial visualization of main differential genes of 3 subclones. (C) Spatial visualization of SNV and RNA clustering for tumor subclones.

**Supplementary Fig. S6.** Characteristics of SNVs at tumor margins. (A) Violin plot illustrating the distribution differences of SNV types across cluster regions in various sections. (B) Spatial visualization of *SPARC* and *GPNMB* gene expression and the distribution of corresponding representative SNVs. (C) Scatterplot showing the correlation between GSVA scores for inflammatory response and hypoxia with normalized SNV counts on spatial spots, including the Pearson correlation coefficient ($R$).

**Supplementary Fig. S7.** SNV group reveals the tumor microenvironment. (A) Spatial visualization of SNV group 0, including the top 8 SNVs ranked by Moran's index. (B) Spatial visualization highlighting SNV group associated with tumor-adjacent regions in the CRC-P19-T. (C) GO term analysis of genes within the SNV windows of SNV group 1 in CRC-P19-T. (D) Gene set variation analysis illustrating the enriched pathways for genes within all SNV windows of SNV group 1 of CRC-P19-T. (E) Network graph showing the connectivity among SNV windows in SNV group 1 of CRC-P19-T, with points representing SNV windows and line lengths indicating the degree of correlation. (F) Spatial visualization of the spatial distribution of highly connected SNV windows (top) and the representative genes contained within these windows (bottom). (G) Spatial visualization of markers associated with B cells in the CRC-P19-T section.

**Supplementary Fig. S8.** Predicting neoantigens from spatial SNVs. (A) Sankey diagram illustrating the differences in mutated and reference base occurrences across various cluster regions. (B) Heatmap generated by inferCNV displaying inferred CNA profiles for mutated and nonmutated spots of S100A11 in the CRC-P59-T2 sample. (C) Single-cell RNA-seq analysis from GSE200997. Top: UMAP projection showing different clusters and epithelial cell distributions. Bottom: Both UMAP projection and violin plot highlighting the specific expression of S100A11 in tumor cells.

**Supplementary Table S1.** Potential neoantigens from the 3 CRC samples.
**Supplementary Table S2.** HLA genotyping prediction.

## Abbreviations

CNVs: copy number variants; COAD: colon adenocarcinoma; CRC: colorectal cancer; DCIS: ductal carcinoma in situ; GO: Gene Ontology; HLAs: human leukocyte antigens; LIHC: liver hepatocellular carcinoma; SNVs: single-nucleotide variants; SV: structural variation; UMI: unique molecular identifiers; UTR: untranslated region.

## Acknowledgments

We thank DCS Cloud (https://cloud.stomics.tech/) for providing the computational resources and software support necessary for this study. This work was jointly supported by the Key Laboratory of Spatial Omics of Zhejiang Province.

## Author Contributions

Y. Li conceived the idea. Y. Li supervised the work with the help of Y. Feng and X. Dong. Y. Li collected the public data with the help of the Y. Li, F. Zhu, and X. Li. Y. Liu performed the data analysis with the help of Y. Li, F. Zhu, and X. Guan. Y. Liu and Y. Li wrote the manuscript with the help of Y. Feng and X. Dong.

## Funding

This research received no specific grant from any funding agency in the public, commercial, or not-for-profit sectors.

## Data Availability

Public spatial transcriptomics data utilized for analysis on the Visium platform were sourced from the Gene Expression Omnibus (GEO, accession GSE181254). Spatial transcriptomics and DNA data on the Slide-seq platform were obtained from the Sequence Read Archive (SRA, accession PRJNA768453). Liver cancer spatial transcriptomics data on the Stereo-seq platform were acquired from the China National GeneBank (CNGB) Sequence Archive (CNSA, accession code: CNP0002199). The spatial transcriptomics data of colorectal cancer obtained through Stereo-seq were sourced from the Genome Sequence Archive (GSA, accession number: PRJCA020107) and the China National GeneBank DataBase (CNSA, accession code: CNP0002432), while the scRNA-seq data were derived from GSE200997. Other data further supporting this work are openly available in the *GigaScience* database, GigaDB [85].

## Competing Interests

The authors declare that they have no competing interests.

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
