## [Peer Review File · GigaScience]

Author's Response To Reviewer Comments

Dear Editor,

We sincerely appreciate the attention and valuable assistance provided regarding our manuscript. In response to the comments and requirements, the following revisions have been made:

1. The manuscript format has been thoroughly revised to align with the specified guidelines.
2. In the Methods section, URLs have been replaced with RRIDs to standardize the citation format, ensuring better compliance with journal standards.
3. Updated figures have been provided, incorporating all necessary modifications and improvements. For example, we have updated Figure 1 panels B, D, E, and F; Figure 4 panels I and J; and Figure 6 panels A-D.
4. Additionally, the GitHub repository in the RRID is <https://github.com/ly4dsg/SpatialSNV>. We will add a redirect link to <https://github.com/YoungLi88/SpatialSNV/> in this GitHub repository. Meanwhile, we are applying via email to update the GitHub website in the RRID.

We believe these revisions have significantly enhanced the quality and clarity of the manuscript. We are grateful for the opportunity to improve our work and hope that the revised version meets the journal's standards for publication.

Thank you for your time and consideration. Sincerely yours,
Young Li on behalf of all authors
Young Li, Ph.D., BGI Research, Hangzhou 310030, China.
Email: liyong13@genomics.cn

Editorial
Manager[®]

Cookie Preference Center

We use cookies which are necessary to make our site work. We may also use additional cookies to analyze, improve and personalize our content and your digital experience. For more information, see our Cookie Policy.

You may choose not to allow some types of cookies. However, blocking some types may impact your experience of our site and the services we are able to offer. See the different category headings below to find out more or change your settings.

Allow all
Manage Consent Preferences

Strictly Necessary Cookies

Always active

These cookies are necessary for the website to function and cannot be switched off in our systems. They are usually only set in response to actions made by you which amount to a request for services, such as setting your privacy preferences, logging in or filling in forms. You can set your browser to block or alert you about these cookies, but some parts of the site will not then work. These cookies do not store any personally identifiable information.

Functional Cookies

These cookies enable the website to provide enhanced functionality and personalisation. They may be set by us or by third party providers whose services we have added to our pages. If you do not allow these cookies then some or all of these services may not function properly.

Performance Cookies

These cookies allow us to count visits and traffic sources so we can measure and improve the performance of our site. They help us to know which pages are the most and least popular and see how visitors move around the site.

Targeting Cookies

These cookies may be set through our site by our advertising partners. They may be used by those companies to build a profile of your interests and show you relevant adverts on other sites. If you do not allow these cookies, you will experience less targeted advertising.

Cookie List

Clear

Apply Cancel

Consent Leg.Interest

Confirm my choices